# Molecular and Serological Detection of Piroplasms in Horses from Nigeria

**DOI:** 10.3390/pathogens10050508

**Published:** 2021-04-23

**Authors:** Idoko S. Idoko, Richard E. Edeh, Andrew M. Adamu, Salamatu Machunga-Mambula, Oluyinka O. Okubanjo, Emmanuel O. Balogun, Sani Adamu, Wendell Johnson, Lowell Kappmeyer, Michelle Mousel, Massaro W. Ueti

**Affiliations:** 1Department of Veterinary Pathology, Faculty of Veterinary Medicine, University of Abuja, Gwagwalada 902101, Nigeria; idoko.sunday@uniabuja.edu.ng; 2Department of Veterinary Medicine, Surgery and Radiology, Faculty of Veterinary Medicine, University of Jos, Jos 930222, Nigeria; edehe@unijos.edu.ng; 3Department of Veterinary Public Health and Preventive Medicine, Faculty of Veterinary Medicine, University of Abuja, Gwagwalada 902101, Nigeria; andrew.adamu@uniabuja.edu.ng; 4Department of Microbiology, Faculty of Sciences, University of Abuja, Gwagwalada 902101, Nigeria; salamatu.mambula@uniabuja.edu.ng; 5Department of Veterinary Parasitology and Entomology, Faculty of Veterinary Medicine, Ahmadu Bello University, Zaria 810107, Nigeria; sookubanjo@abu.edu.ng; 6Department of Biochemistry, Faculty of Life Sciences, Ahmadu Bello University, Zaria 810107, Nigeria; eobalogun@abu.edu.ng; 7Department of Veterinary Pathology, Faculty of Veterinary Medicine, Ahmadu Bello University, Zaria 810107, Nigeria; saniadamu@abu.edu.ng; 8Animal Disease Research Unit, USDA-ARS, Pullman, WA 99164-6630, USA; carl.johnson@usda.gov (W.J.); lowell.kappmeyer@usda.gov (L.K.); michelle.mousel@usda.gov (M.M.); 9Department of Veterinary Microbiology & Pathology, College of Veterinary Medicine, Washington State University, Pullman, WA 99164-7040, USA

**Keywords:** equine piroplasmosis, horses, diagnostic assays, ixodid ticks

## Abstract

Equine piroplasmosis, an economically important disease of equids caused by the hemoprotozoan parasites *Theileria equi*, *T. haneyi*, and *Babesia caballi*, has a worldwide distribution. These parasites are transmitted by ixodid ticks. To improve the detection of horses in Nigeria exposed to piroplasm parasites, 72 horses with variable clinical signs of piroplasmosis were sampled from Northwest and Northcentral Nigeria and tested by nPCR and cELISA. Blood and serum samples were collected from each horse via jugular venesection. Individually, nPCR or cELISA failed to identify all horses exposed to piroplasms. A combination of species-specific nPCR and the OIE-approved *T. equi* and *B. caballi* cELISAs enhanced the detection of horses exposed to parasites. The results also demonstrated horses showing abnormal hematology were positive for only *T. equi,* except for one sample that was coinfected with *T. equi* and *T. haneyi*. We also identified ticks collected from some of the horses, with *Rhipicephalus evertsi evertsi* being the most prevalent. This study shows that a larger proportion of horses in the sample set were exposed to *T. equi* than *B. caballi* or *T. haneyi*. Additionally, ticks that have been previously reported as potential vectors for these parasites were found to have infested sampled horses. Further studies are needed to investigate which tick species are competent vectors for *Theileria* spp. and *Babesia caballi* in Nigeria.

## 1. Introduction

Equine piroplasmosis (EP) is an economically important disease of equids caused by the hemoprotozoan parasites *Theileria equi* and *Babesia caballi* [1], and the recently defined *T. haneyi* [2]. In Nigeria, economic benefits arise from the use of horses in ceremonies (durbars and parades), recreation, agriculture, food, and as companion animals [3]. Those economic benefits are challenged by numerous problems, including disease, poor management practices, malnutrition, and unrestricted cross-border movement of horses [4]. All these factors contribute to the maintenance and spread of EP parasites but may be mitigated by a better understanding of parasite distribution within regions of this large, ecologically, and culturally diverse country.

Additionally, EP parasites are directly associated with the presence and distribution of their vectors [5]. These parasites are transmitted naturally by ixodid ticks, including *Rhipicephalus* [6,7], *Dermacentor* [8], *Amblyomma* [5,9], *Haemaphysalis* [10], and *Hyalomma* [11]. The transmission *Theileria* spp by ticks may be transstadial or intrastadial [10,12]. Transstadial transmission occurs when a tick stage acquires the pathogen from an infected horse, and the next life cycle stage within the same tick generation transmits the pathogen, whereas intrastadial transmission is when the same tick stage that acquired the parasite transmits the pathogen [7,12]. In contrast, the transovarial transmission of *B. caballi* occurs when the parasite is transmitted by the next tick generation [8]. In utero transmission rarely occurs [13,14,15], and iatrogenic transmission via the use of blood-contaminated equipment or needles and transfusion of infected blood is well documented [9,16,17]. After transmission, horses suffer acute infection showing high fever, anemia, jaundice, and lethargy [17]. *Theileria equi* is the most virulent among the horse piroplasms; horses become persistently infected with piroplasms for life and are reservoirs for pathogen transmission [12]. Drug treatment has been shown to eliminate parasites efficiently [18,19] and reduce transmission risk. However, antibodies remained detectable for months after parasite elimination [19].

Equine piroplasmosis is endemic and has long been documented in Nigeria. The epidemiological data for EP in Nigeria, as reviewed by Onyiche et al. (2020), were based on low sensitivity assays, such as microscopic examination of a stained blood smear or a single assay, including PCR or competitive ELISA (cELISA) [20]. In the case of PCR, detection of the parasite can precede the development of a detectible antibody response. In the case of cELISA, after drug therapy, horses remain serologically positive for months [19]. Therefore, there is a need for combined high sensitivity molecular and serological studies that enhance the detection of animals exposed to the parasites that cause EP in Nigeria. Only in the last year has the use of molecular tools been reported in Nigeria [4,21,22]. Here, we report using the World Organization for Animal Health (OIE) approved cELISA serological tests [23] combined with molecular detection for *T. equi*, *T. haneyi*, and *B. caballi* to increase our ability to detect horses exposed to pathogens that cause EP. 

Further, the association of EP parasites with competent tick vectors in specific districts will contribute to the understanding of risk levels for uninfected horses. Targeting symptomatic animals, while not helpful in describing prevalence, will help make associations with tick presence and disease symptoms. In this study, blood samples from symptomatic horses from Northwest and Northcentral Nigeria were evaluated for exposure to EP parasites by combining molecular and serological assays, and they were associated with tick species present on individual horses.

## 2. Results

Before sample collection, the 72 horses examined in this study showed variable clinical signs. However, microscopic examination of Giemsa-stained thin blood smear did not detect any infected erythrocytes. Red blood cell (RBC) count, packed cell volume (PCV), and hemoglobin showed 41 horses were below the normal range for at least one out of the three parameters (Table 1, Appendix A). 

Evaluation of 72 horses by nPCR or cELISA demonstrated 88.88% (64/72) of horses had been exposed to EP parasites (Table 1 and Appendix A). Eighteen horses were both T. equi nPCR and cELISA positive. Forty-three horses were T. equi cELISA positive and nPCR negative. Two horses were nPCR positive and cELISA negative (Table 2). In addition, one T. equi positive horse sample was also positive for B. caballi (z-1) (Appendix A). Two horse samples positive for T. equi were also positive for T. haneyi (z-42 and z-46) (Appendix A). One horse sample was only T. haneyi nPCR positive (z-54) (Appendix A). Nineteen horses had RBC count, PCV, and hemoglobin below the normal ranges. Of the 19, 18 samples were positive for T. equi, and one horse sample, z-42, was positive for T. equi and T. haneyi. Eleven horses had two of three parameters below the normal ranges. Nine samples were positive for only T. equi, sample z-1 was positive for T. equi and B. caballi, and z-26 was negative for all tests. Eleven horses had two of three parameters below the normal ranges, of which nine samples were positive for T. equi and two samples were negative for all tests (Appendix A).

In this study, we identified ixodid tick species feeding on horses in Nigeria. Of 444 collected ticks, 51.1% and 48.9% were identified as male and female ticks, respectively (Table 3 and Appendix A). A few of the tick species identified in this study have been implicated in previous studies for transmitting pathogens that cause equine piroplasmosis, including *Rhipicephalus evertsi evertsi*, *R. sanguineus*, *Hyalomma dromedarii*, and *H. truncatum* [4].

Binomial categorization of nPCR for *T. equi* showed that horses from Zaria were significantly more infected (*p* < 0.007) compared to horses from Igabi or Keffi. We did not find any association between *T. equi* infection, as detected by nPCR, and horse sex (*p* > 0.84). Competitive ELISA results highlighted that mares and horses from Zaria were significantly more positive for *T. equi* (*p* < 0.001). Female horses were significantly more likely to be detected as infected with *T. equi* than males (*p* < 0.02) when both testing methods were analyzed together. Overall, horses from Zaria were significantly more infected (*p* < 0.001) than horses from other locations sampled. No differences between horse sex or location were detected for *T. haneyi* (sex: *p* > 0.18; location: *p* > 0.70) or *B. caballi* (sex: *p* =1; location: *p* > 0.79) parasites. We did not demonstrate significant differences between the two detection methods employed in this study (i.e., nPCR and cELISA) for the tick species identified on horses for the parasite species tested (*p* > 0.45). No ticks were found on horses that tested positive for *T. haneyi*. Moreover, we did not detect any significant differences in parasite prevalence in the three tested locations (*p* > 0.34) when tick species were included in the model. Sex could not be tested as there were only males in Zaria, where most of the horses sampled had ticks. The number of ticks per horse was higher in Zaria for all testing methods (*p* < 0.02) and strongly associated with *T. equi* nPCR (*p* < 0.001) in positive animals having more ticks. Competitive ELISA of *T. equi* was not associated with tick number (*p* > 0.46). Sex was not significant for the number of ticks for any testing method (*p* > 0.77).

## 3. Discussion

Equine piroplasmosis is spread worldwide, including Africa, the Americas, Asia, Australia, and Europe. However, a few countries are considered to have equine piroplasmosis-free status [17]. Nevertheless, the movement of live horses between countries could widely disseminate new parasite variants [26]. Understanding the infection status of horses may assist in preventing parasites from spreading into different geographical regions. This study’s focus was to determine the exposure status of horses showing clinical signs of EP in Northwest and Northcentral Nigeria by molecular and serological assays. The results demonstrated that most horses showing clinical signs suggestive of piroplasmosis were exposed to *T. equi*. The result also showed a predominant infestation of *R. evertsi evertsi*, a vector implicated for transmission of parasites that cause EP [5]. However, this study was limited in concluding the effect of horse breed and sex due to the unbalanced representation of breed and sex in the three counties. Future work should consider an experimental design with increased sample size and a more balanced representation of horse breed and sex across geographical locations being sampled. 

Although almost half of the horses sampled in this study presented variable clinical signs consistent with piroplasmosis, parasitemia was below the detection limit by micro-scopic examination of Giemsa-stained blood film. Motloang et al. (2008) did not detect *B. caballi* or *T. equi* in the blood smears they investigated from EP suspected horses but recorded high seroprevalence of the parasites using immune fluorescence antibody tests [27]. Moloi et al. (2010) did not detect any parasites using thin blood smears from 543 horses screened for EP [28]. The mere absence of positive results by microscopic examination does not indicate that horses are not infected with EP parasites. It suggests difficulty detecting *B. caballi* and *T. equi* at the onset of acute disease and in carrier animals with low parasitemia [29,30]. During acute parasitemia, nPCR detects infection before an antibody response develops, yielding a negative cELISA result. In chronic infection, the low parasitemia stimulates antibody production for life. This again justifies the use of combined molecular and serological assays to overcome the constraint of single assays. 

The current study documented evidence of anti-T. equi antibodies to the epitope defined by the mAb 36/133.97 [31] in horse samples from Nigeria. In this study, *T. equi* detection was highest using cELISA rather than species-specific nPCR. This finding corroborates that of Ibrahim et al. (2011) in horse samples from Egypt. The prevalence of *T. equi* infection using cELISA was higher, followed by PCR, and then microscope examination of Giemsa-stained blood film [32]. Possible explanations for the lower nPCR detection are a low number of parasites in the peripheral blood of carrier state animals or horses having received treatment to eliminate or reduce parasitemia. Both scenarios may result in antibody detection with nPCR being negative as previously described [1].

Individually, nPCR or cELISA failed to identify all horses infected with piroplasms and emphasized the need for combined assays to detect horses exposed to EP parasites. We demonstrated that ~85% of samples were positive for *T. equi* (21 samples nPCR positive and 61 cELISA positive), which was higher than that for *B. caballi* (1 sample nPCR positive). These findings agree with previous reports [33,34]. A possible explanation for the low detection of *B. caballi* in this study may be associated with genetic diversity between the rap-1 genes of the Nigerian *B. caballi* isolate and the isolate used for the currently available RAP-1-cELISA and rap-1-PCR. Recent epidemiological evidence reported discordance in the sequence conservation between *B. caballi* isolates from Egypt [35], Israel [36], South Africa [37], and Spain [38] and the epitope used as the antibody recognition site in the OIE-approved *B. caballi* RAP-1-cELISA [23,39]. 

A few horses in this study were positive for *T. haneyi* by nPCR. Unfortunately, there are no commercial cELISA or IFAT available to detect antibodies to *T. haneyi*. Recently, a *T. haneyi* ELISA has been developed and should be included in future epidemiological studies [40]. Based on our findings using available nPCR and cELISA in this study, we expect serological assays to reveal a different outcome regarding horses exposed to *T. haneyi* in Nigeria. Considering that some horses may be exposed to different variants, in this study, we may have missed horses that were exposed to *T. equi*-like variant strains [26]. Therefore, additional research concerning different variants, if not species of parasites, that infect horses is needed. Recently, a Theileria species 18S rDNA variant was elevated as a distinct species [2]. Recent investigations using phylogenetic analysis of rDNA showed Theileria piroplasm variants from horses in Nigeria, including *T. haneyi* [21,41].

In this study, we also identified ticks that are potential vectors for EP parasites feeding on horses showing signs of disease. Previous studies demonstrated that *R. evertsi evertsi*, *R. sanguineus*, and *H. dromedarii* are vectors for *T. equi* and *B. caballi* [4]. Hyalomma truncatum is a vector of *B. caballi* [42]. However, there are no studies concerning vector competency for *T. haneyi*. Therefore, transmission studies to confirm the tick competence of *T. equi*, *B. caballi*, and *T. haneyi* in Nigeria will be critical for the development of control strategies to prevent the spread of these parasites in Nigerian horse populations.

## 4. Materials and Methods

### 4.1. Study Population

A purposive sampling technique using 72 horses of West African Barb or Argentine Polo Pony breeds and sexes (Figure 1) showing varying clinical signs suggestive of piroplasmosis was utilized for this study. Clinical signs included congested to icteric ocular and vaginal mucous membranes, pyrexia, edema around the fetlock, weight loss, anorexia, and unilateral and bilateral epiphora. Blood and serum samples were collected from horses in Zaria and Igabi, the Local Government Areas of Kaduna State, and the Keffi Local Government Area of Nasarawa State (Figure 1). 

The consent of the client and the attending veterinarian was sought before sample collection. Three milliliters of whole blood were dispensed into vacutainers containing ethylenediaminetetraacetic acid (EDTA), out of which 200 µL was spotted on Whatman FTA^TM^ classic cards (Whatman® International Ltd., Hercules, CA, USA) and allowed to dry at room temperature. Complete blood counts were performed for all 72 horses by the College of Veterinary Medicine, University of Abuja, Nigeria, using an automated hematology analyzer (Pioway Medical Lab Equipment Co., Ltd., Nanjing, China). A few drops of EDTA blood from the respective horses were used to perform Giemsa-stained thin blood smear on glass slides and 60 fields viewed using light microscopy at 100x oil immersion magnifications. Another 5 mL of whole blood was dispensed into plain vacutainers and allowed to stand at room temperature for 1 h, after which sera were harvested and transferred into 2 mL Eppendorf tubes and stored at −4 °C until used. 

### 4.2. Polymerase Chain Reaction (PCR)

One-millimeter disposable biopsy punches with a plunger (Miltex, Inc. Princeton, NJ, USA) were used to punch out discs from each card into labeled PCR amplification tubes. Discs were washed three times with 200 µL of FTA card purification reagent added into each PCR tube and incubated at room temperature for 5 min. Two hundred microliters of Dulbecco’s phosphate buffered saline (DPBS) (Life Technologies Corporation, Grand Island, NY, USA) were added to each PCR tube and incubated at room temperature for 5 min. The DPBS was removed, and the step repeated one more time. The lid of each PCR tube was opened, and the disc was allowed to dry for 1 h at room temperature. 

Modified nPCRs for *T. equi*, *B. caballi*, and *T. haneyi* were performed as previously described [2,8]. The primer sets are presented in Table 4. The first-round PCR reaction was performed in 25 µL reaction volumes with 12.5 µL of PCR master mix (Roche, Indianapolis, IN, USA), 7.5 µL of external primer mix, and 5 µL of water containing the sample disc. Amplification was conducted using a Bio-Rad 1000^TM^ Thermal cycler (Bio-Rad Laboratories, Inc., Summit, MS, USA); cycling conditions were 95 °C for 5 min, followed by 30 cycles of 95 °C for 20 s, 60 °C for 20 s, and 72 °C for 20 s, and then a final extension of 72 °C for 5 min, and held at 4 °C. Second-round reactions contained 1 µL of the first-reaction product, 12.5 µL of PCR master mix (Roche, Indianapolis, IN, USA), and 11.5 µL of internal primer mix. The cycling conditions were 95 °C for 5 min, followed by 30 cycles of 95 °C for 5 s, 60 °C for 5 s, and 72 °C for 5 s, a final 72 °C for 5 min extension, and held at 4 °C. The PCR products were separated on 2% agarose gel (SeaKem LE, Lonza, Rockland, ME, USA) in 1× TAE (0.04 M Tris, 0.004 M acetate, 0.001 EDTA, pH 8.3). Agarose gels were electrophoresed for 1 h at 70 volts and stained with SYBR Safe DNA stain according to manufacturer directions (ThermoFisher Scientific, Waltham, MA, USA).

### 4.3. Detection of T. equi- and B. caballi-Specific Antibodies

Serological testing was carried out using a competitive Enzyme-linked Immunosorbent Assay (cELISA) kit (VMRD, Pullman, WA, USA) as directed by the manufacturer, with ≥40% inhibition as the definition of a positive reaction, per the manufacturer protocol. All the samples were run in duplicates. There is no commercially available ELISA test to detect *T. haneyi*-specific antibodies.

### 4.4. Morphological Identification of Ticks

Ticks collected from the horses were transferred into Petri dishes, mounted on slides, and examined using a dissecting microscope at a magnification of 40× and sorted by genus, species, and sex. Ticks were morphologically identified using features described by Walker et al. (2014) [43].

### 4.5. Statistical Analysis

Data were analyzed with a general linear model in SAS 9.4 with fixed effects of county and sex for response variables of the testing method. Sex was categorized as male and female to reduce confounding within counties. The breed of horse was not included in the analysis as West African Barb horses were confounded in one county. To test if tick species impacted detection of parasites, data were analyzed with a general linear model with fixed effects of tick species and county. Sex was not included in this model as Zaria had the most identified ticks and only male horses. The number of ticks on horses was tested with a general linear model with fixed effects of sex, county, and parasite testing method. All *p*-values < 0.05 were considered significant.

## 5. Conclusions

A combination of molecular and serologic tests improved our ability to detect horses exposed to EP pathogens in Nigeria. Both West African Barb horses and the Argentine Polo Pony breeds had high exposure rates, which may be associated with poor management practices among the locals. Hematology and microscopic examination proved to be unreliable in determining piroplasm exposure. Therefore, to prevent spreading parasites that cause EP in Nigeria, continuous surveillance using a combination of molecular and serologic assays is critically important. 

## Figures and Tables

**Figure 1 pathogens-10-00508-f001:**
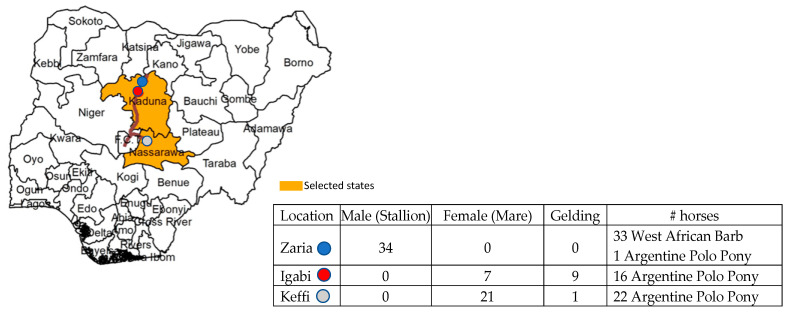
Site of sampling in Nigeria and category of horses utilized in this study.

**Table 1 pathogens-10-00508-t001:** Summary of hematology, molecular, and serological data of horse samples collected in Nigeria.

Piroplasm							
		*T. equi*		*B. caballi*		*T. haneyi*	
Hematology	Horses tested	nPCR	cELISA	nPCR	cELISA	nPCR	cELISA
Abnormal	41	15	37	1	0	1	NA
Normal	31	5	24	0	0	2	NA
Total	72	20	61	1	0	3	NA

RBC: red blood cells (normal range: 6.6–10^6^/µL); PCV: packed cell volume (normal range: 31%–50%); Hb: hemoglobin (normal range: 11.4–17.3 g/dL) [24,25]; Abnormal: samples below normal range for at least one parameter; nPCR: nested PCR; cELISA: competitive ELISA; NA: not commercially available.

**Table 2 pathogens-10-00508-t002:** Comparison of molecular and serologic assays to detect T. equi and B. caballi in horses.

	Piroplasm	
nPCR/cELISA	*T. equi*	*B. caballi*
+/−	2	1
−/+	43	0
+/+	18	0
Total	63	1

nPCR: nested PCR; cELISA: competitive ELISA.

**Table 3 pathogens-10-00508-t003:** Distribution of various tick species infesting Nigerian horses.

Tick Species	Male	Females	Total
*Rhipicephalus evertsi evertsi*	164 (50.8%)	159 (49.2%)	323 (69.6%)
*Rhipicephalus sanguineus*	4 (40.0%)	6 (60.0%)	10 (2.2%)
*Amblyomma variegatum*	37 (55.2%)	30 (44.8%)	67 (14.4%)
*Hyalomma dromedarii*	22 (68.8%)	10 (31.2%)	32 (6.9%)
*Hyalomma impeltatum*	0	4 (100%)	4 (0.9%)
*Hyalomma truncatum*	0	1 (100%)	1 (0.2%)
*Boophilus decoloratus*	0	7 (100%)	7 (1.5%)
Total	227 (51.1%)	217 (48.9%)	444 (100%)

**Table 4 pathogens-10-00508-t004:** Species-specific nested polymerase chain reaction.

Primer	Sequence (5’-3’)	Expected Product Size	Target Gene
*T. equi* external-forward	GAGGAGGAGAAACCCAAG	549 bp	*ema1*
*T. equi* external-reverse	GCCATCGCCCTTGTAGAG		
*T. equi* internal-forward	TCAAGGACAACAAGCCATAC	229 bp	
*T. equi* internal-reverse	TTGCCTGGAGCCTTGAAG		
*B. caballi* external-forward	GATTACTTGTCGGCTGTGTCT	374 bp	*rap1*
*B. caballi* external-reverse	CGCAAAGTTCTCAATGTCAG		
*B. caballi* internal-forward	GCTAAGTACCAACCGCTGA	221 bp	
*B. caballi* internal-reverse	CGCAAAGTTCTCAATGTCAG		
*T. haneyi* external-forward	CCATACAACCCACTAGAG	381 bp	hypothetical single copy gene
*T. haneyi* external-reverse	CTGTCATTTGGGTTTGATAG		
*T. haneyi* internal-forward	GACAACAGAGAGGTGATT	238 bp	
*T. haneyi* internal-reverse	CGTTGAATGTAATGGGAAC		

## Data Availability

Data are contained within the article and Appendix A. Additional raw data are available upon request to the corresponding author.

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
