# Peer review of "Molecular and Serological Detection of Piroplasms in Horses from Nigeria"

_pathogens, 2021, doi:10.3390/pathogens10050508_

Round 1
Reviewer 1 Report
This is an interesting study dealing with the occurrence of Babesia/Theileria in symptomatic horses from Nigeria, along with the identification of ticks. The data are of interest for the readership.
Some suggestions are listed below
Title of paper - please, modify in "Molecular and serological detection of piroplasms in horses from Nigeria"
line 56 . the sentence is referred toTheileria equi, however a recent paper (doi: 10.3390/ani10020341) consider this route as uncommon.
Please check T.equi and other proper names are written in italics (i.e. lines 86 and following, 96 and following..., line 111..and quite all discussion and materials and methods).
The Authors should merge tables 3 and 4 (they should be 1 and 2, please check the numbers of tables throughout the text)
material and methods
Some anamnestic data about the selected horses should be provided. Were they submitted to acaricide treatments? If so, please specify. Was the number of ticks per horse extimated?
A pcr study to evaluate the occurrence of piroplasm DNA in ticks would be of interest to complete data about the potential of such vectors.
Author Response
Dear Editor
We appreciate the comments and have responded point-by-point to each comment with changes to the manuscript.
Reviewer #1.
Title of paper - please, modify in "Molecular and serological detection of piroplasms in horses from Nigeria"
Response: Modified as recommended.
line 56 . the sentence is referred toTheileria equi, however a recent paper (doi: 10.3390/ani10020341) consider this route as uncommon.
Response: In that manuscript, T equi does not appear to be a major cause for abortion. However, transplacental transmission does rarely occur. We modified the statement and added the suggested reference.
Line 62-64: “In utero transmission rarely occurs [13, 14, 15], and iatrogenic transmission via the use of blood-contaminated equipment or needles and transfusion of infected blood is well doc-umented [9, 16, 17].”
Please check T.equi and other proper names are written in italics (i.e. lines 86 and following, 96 and following..., line 111..and quite all discussion and materials and methods).
Response: We fixed all the italics as recommended.
The Authors should merge tables 3 and 4 (they should be 1 and 2, please check the numbers of tables throughout the text)
Response: We merged tables 3 and 4 as recommended.
material and methods
Some anamnestic data about the selected horses should be provided. Were they submitted to acaricide treatments? If so, please specify. Was the number of ticks per horse extimated?
Response: Unfortunately, we did not collect acaricide treatment data. We added a table as supplementary data for tick estimation.
A pcr study to evaluate the occurrence of piroplasm DNA in ticks would be of interest to complete data about the potential of such vectors.
Response: In our understanding, ticks that are positive for the parasites does not mean it is a competent vector. During tick feeding, the parasites are ingested, and we would expect PCR positives. Now that we know which ticks are feeding on horses in Nigeria, the next experiments are to rear uninfected ticks in the laboratory and test if these ticks can acquire and transmit the parasite to naïve horses.
Reviewer 2 Report
In this study, Dr. Idoko and colleagues have highlighted the importance of using a combination of molecular and serological test to improve the ability to detect equine piroplasmosis pathogens in horses in Nigeria. In their conclusions, the authors emphasized on the importance of conducting a continuous surveillance using these highly sensitive methods to understand parasite transmission and therefore, prevent spreading of EP parasites. Overall, the manuscript is well written and easy to read and understand and the data collected are highly valuable. However, the authors really have to improve the presentation of their data (graphs instead of big raw data tables), statistically analyze and compare their groups (see details below) and integrate their findings to the existing literature. In the present state, data are not statistically analyzed and compared and this is a major flaw of the study. Also, the authors should provide more details about their protocols: especially how they determine physiological parameters like RBC count, PCV, and hemoglobin rate; or the reasons to choose these specific parasite target genes for their PCR assays. All these info should be added and described in the methods section. Finally, I found the tick collection and identification really valuable but I don't understand why the authors did not perform molecular test to check the potential infection of the vectors with EP parasites. These data could provide important information about which species of ticks are vector of which species of parasite, see if the authors observed differences in the prevalence of infection between male ad female ticks for a given species, if some species can be infected by multiple parasite species. All these points (and many others, see below) need to be addressed before considering the acceptance of the manuscript for publication.
Decision: Major revision
======= LEGENDS =======
p.: page
§: paragraph
l.: line
Fig.: figure
Tab.: table
> : replaced by
======= MAJOR COMMENTS =======
------- COMMENTS ON THE TEXT -------
- Major comments -
*Introduction:
- l.53: Could the authors please define and explain more clearly the transmission of the pathogens by ticks: what is exactly an intrastadial, transtadial and transovarial transmission?
- l.60: The authors mentioned "all these studies" but did not provide references for them. Could the authors add these particular refs in their present manuscript?
- l.60: What are the low sensitivity assays or single assay mentioned here? What is method used? Only microscopy? The authors should described more clearly what is currently use for detecting EP in horses.
- l.65: What is OIE? Please define the acronym.
- l.65: What is the difference between cELISA and ELISA? Please explain briefly.
The authors never mentioned how horses are treated (which drug, percentage of success/failure of EP treatment?) after have been diagnose with EP or if some EP pathogen species are more virulent to horses. These info should be provided by the authors in the introduction.
*Results:
- l.80: Please define the acronyms (RBC, PCV) when first cited in the main text.
- Table 3: The header of the samples column is somewhat confusing. Please put "Sample ID" where "Normal range" is written and move "Normal range" elsewhere (closer to the values of RBC, PCV and Hb).
- The table 3 is very long and not very useful for the reader (at least in the main text of the manuscript). I strongly suggest to the authors to put all the detailed tables as supplementary files and summarize the info present in the table into graphs.
Authors can use several representation for their data:
- A barplot for each of the 3 parameters you have tested (RBC, PCV and Hb). Each bar represent a sample. Authors can draw lines to indicate the lower and upper limits of the normal range for each of the physiological parameters measured.
- A beeswarm plot for each variable where each sample is represented by a dot. Authors can color sample points which exhibit values below the normal range.
Graphs are far more visual than table and easier to interpret. The authors can use R software (free) to represent your data and to perform their statistical analyses.
- I also suggest to the authors to provide a sampling map as Figure for their manuscript.
- Table 4: what does the % PI mean for T. equi cELISA and B. caballi cELISA? These % are somewhat confusing because they are sometimes negative (e.g. Sample 26, Z-9, Z-10 etc.). The authors should explain how and why they have obtained these negative values.
Again, this table 4 is very big and difficult to read or interpret. I would advice authors to put this very detailed table in Supplementary files and present a graph to summarize their data, as suggested above for Table 3.
These kind of data could be summarized and represented as a pie chart or a cumulative barplot, with the percentage of horses respectively infected with either T. equi, T haneyi and B. cabali and the % of horses infected with 2 parasite species.
Also if the authors prefer to have a table instead of a graph, I strongly encourage them to present a
summarized version of this table 4, with the total number and percentage of horses infected with each parasites for nPCR and for cELISA and the total nb and percentage of bi-infected horses. With this summary table, authors should also be able to statistically compare the sensitivity of both methods employed (i.e. nPCR and cELISA) for T. equi and B. caballi.
The authors should also integrate in their analysis the location of their samples and the horses breed and sex. Also, authors should perform a statistical analysis of their dataset. This is crucial to identify high transmission areas or if some horse sex or breed are more prone to infection, and it is really lacking in this study. Generalized linear model can be used here, to see if geographical location, horse breed, horse sex, can significantly influence the presence of some tick species and also the infection with the different pathogens identified.
- Why the authors failed to get positive results with cELISA for B. caballi for sample z-1, where nPCR test is positive? Too low prevalence of parasite in the blood for getting enough antibody for an ELISA test? Do the ELISA tests were run in duplicates to be sure about the result found for each sample?
- l.96 and Table 5: Did the authors test the presence of parasites (using the nPCR assay) in the ticks collected to check their infection status? It would have been interesting to see the prevalence of the different parasites in the different tick species and gender. This is particularly important to understand how the different parasite species are transmitted and which vector is used, or if a vector can carry different parasite species.
*Discussion:
- l.126-127: How long do the anti-T antibodies last in horse blood after an infection?
- l.152-153: Could it be possible to chose very conserved or several conserved genomic regions for
each parasite species to be able to detect also parasite which harbor variation in these genomic regions?
This shouldn't be a problem if you have several molecular markers for each species, designed in conserved regions.
- l. 155-157: Why not having tested these collected ticks for the presence of parasites using the nPCR sets of primers? Could the authors identified which ticks were collected from which horse and add the info in their analysis. Authors could then potentially established correlations between presence of specific tick species and EP parasite in horse blood.
*Materials and Methods:
- Looking at the Table 1, we can see that we clearly have a sex ratio bias for each of the two breeds: could the authors explain why? With such bias, we cannot test for parasite prevalence by sex for each breed but for all the breeds together it is still possible and valuable.
- l.175: Do RBC counts where performed using microscopy only? This is not very clear. Did the authors also spotted parasites while evaluating the level of RBC or never found any parasite?
How PCV was measured?
How hemoglobin rate was assessed?
All these methodological information are missing and should be added to the manuscript.
- l.191: How did the authors choose the targeted genes and what is the potential function of these genes in the EP parasites (if known)? How the primers were designed?
- l.197: Did the authors cleaned their PCR product after the first round PCR to deactivate the first set of primers? (using ExoSAP-it for e.g.).
There are also several information that are missing in this section, for example:
- How the authors separated their PCR products: Agarose gel? What % of agarose?
- What are the electrophoresis conditions used in this experiment?
- What is the expected size of the amplicon for each pair of primers?
Also could the authors provide a gel example of their PCR products to illustrate the kind of electrophoresis gel results for (i) a single infection for each of the EP parasites and (ii) a double infection?
======= MINOR COMMENTS =======
------- COMMENTS ON THE TEXT -------
- Minor comments -
*Results:
- l.88: Please remove the blank space before "In addition" and check in the whole manuscript if more than one blank space is present at the beginning of sentences.
- l.89: Please be sure to always put all the scientific names in italic in the whole manuscript.
*Materials and Methods:
- Authors have to check how they numbered their tables: Table 1 should be first and is coming at the end of the manuscript.
----- COMMENTS ON THE REFERENCES -----
- l.60: The authors mentioned "all these studies" in the introduction but did not provide references for them. Could the authors add these particular refs in their present manuscript?
Author Response
Dear Editor
We appreciate the comments and have responded point-by-point to each comment with changes to the manuscript.
Reviewer #2.
Major comments -
*Introduction:
- l.53: Could the authors please define and explain more clearly the transmission of the pathogens by ticks: what is exactly an intrastadial, transtadial and transovarial transmission?
Response: We clarified as requested.
Lines 58-62: Transstadial transmission occurs when a tick stage acquires the pathogen from an infected horse, and the next life cycle stage within the same tick generation transmits the pathogen, whereas intrastadial transmission is when the same tick stage that acquired the parasite transmits the pathogen [7, 12]. In contrast, the transovarial transmission of B. caballi occurs when the parasite is transmitted by the next tick generation [8].
- l.60: The authors mentioned "all these studies" but did not provide references for them. Could the authors add these particular refs in their present manuscript?
Response: We revised the statements.
Line 71-74: “Equine piroplasmosis is endemic and has long been documented in Nigeria. The epidemiological data for EP in Nigeria, as reviewed by Onyiche et al., 2020, were based on low sensitivity assays, such as microscopic examination of a stained blood smear or a single assay, including PCR or competitive ELISA (cELISA) [20].”
- l.60: What are the low sensitivity assays or single assay mentioned here? What is method used? Only microscopy? The authors should described more clearly what is currently use for detecting EP in horses.
Response: We modified the statements to clarify our points.
Line 71-77: “Equine piroplasmosis is endemic and has long been documented in Nigeria. The epidemiological data for EP in Nigeria, as reviewed by Onyiche et al., 2020, were based on low sensitivity assays, such as microscopic examination of a stained blood smear or a single assay, including PCR or competitive ELISA (cELISA) [20]. In the case of PCR, detec-tion of the parasite can precede the development of a detectible antibody response. In the case of cELISA, after drug therapy, horses remain serologically positive for months [19].
- l.65: What is OIE? Please define the acronym.
Response: We added World Organization for Animal Health (OIE) for clarification.
- l.65: What is the difference between cELISA and ELISA? Please explain briefly.
Response: To increase specificity, competitive ELISA uses monoclonal antibodies raised against T. equi EMA-1 protein or B. caballi RAP-1 protein that compete with horse antibodies. In the case of cELISA, the results are presented as percent inhibition of monoclonal antibody. In contrast, ELISA directly detects horse antibodies against the parasites and the results would be presented as absorbency.
The authors never mentioned how horses are treated (which drug, percentage of success/failure of EP treatment?) after have been diagnose with EP or if some EP pathogen species are more virulent to horses. These info should be provided by the authors in the introduction.
Response: We added statements as recommended.
Lines 65-70: “After transmission, horses suffer acute infection showing high fever, anemia, jaundice, and lethargy [17]. Theileria equi is the most virulent among the horse piroplasms. Horses become persistently infected with piroplasms for life and are reservoirs for pathogen transmission [12]. Drug treatment has been shown to eliminate parasites efficiently [18, 19] and reduce transmission risk. However, antibodies remained detectable for months after parasite elimination [19].”
*Results:
- l.80: Please define the acronyms (RBC, PCV) when first cited in the main text.
Response: We defined the acronyms in the results per reviewer request.
Lines 93-95: “Red blood cell (RBC) count, packed cell volume (PCV), and hemoglobin showed 41 horses were below the normal range for at least one out of the three parameters (Table 1, supplementary data 1).”
- Table 3: The header of the samples column is somewhat confusing. Please put "Sample ID" where "Normal range" is written and move "Normal range" elsewhere (closer to the values of RBC, PCV and Hb).
Response: We modified the table per reviewers suggestion.
- The table 3 is very long and not very useful for the reader (at least in the main text of the manuscript). I strongly suggest to the authors to put all the detailed tables as supplementary files and summarize the info present in the table into graphs.
Authors can use several representation for their data:
- A barplot for each of the 3 parameters you have tested (RBC, PCV and Hb). Each bar represent a sample. Authors can draw lines to indicate the lower and upper limits of the normal range for each of the physiological parameters measured.
- A beeswarm plot for each variable where each sample is represented by a dot. Authors can color sample points which exhibit values below the normal range.
Graphs are far more visual than table and easier to interpret. The authors can use R software (free) to represent your data and to perform their statistical analyses.
- I also suggest to the authors to provide a sampling map as Figure for their manuscript.
- Table 4: what does the % PI mean for T. equi cELISA and B. caballi cELISA? These % are somewhat confusing because they are sometimes negative (e.g. Sample 26, Z-9, Z-10 etc.). The authors should explain how and why they have obtained these negative values.
Response: We modified the tables for clarification per reviewers’ comments. Negative PI values happen when the test sample inhibits less than the negative control in the test kit and is well known phenomena of cELISA. The percent inhibition was removed, and data are reported as positive and negative.
Again, this table 4 is very big and difficult to read or interpret. I would advice authors to put this very detailed table in Supplementary files and present a graph to summarize their data, as suggested above for Table 3.
These kind of data could be summarized and represented as a pie chart or a cumulative barplot, with the percentage of horses respectively infected with either T. equi, T haneyi and B. cabali and the % of horses infected with 2 parasite species.
Also if the authors prefer to have a table instead of a graph, I strongly encourage them to present a
summarized version of this table 4, with the total number and percentage of horses infected with each parasites for nPCR and for cELISA and the total nb and percentage of bi-infected horses. With this summary table, authors should also be able to statistically compare the sensitivity of both methods employed (i.e. nPCR and cELISA) for T. equi and B. caballi.
Response: We modified per reviewer suggestion by summarizing the data in a table format.
The authors should also integrate in their analysis the location of their samples and the horses breed and sex. Also, authors should perform a statistical analysis of their dataset. This is crucial to identify high transmission areas or if some horse sex or breed are more prone to infection, and it is really lacking in this study. Generalized linear model can be used here, to see if geographical location, horse breed, horse sex, can significantly influence the presence of some tick species and also the infection with the different pathogens identified.
Response: We performed statistical analysis as recommended.
Line 141-154: “ Binomial categorization of nPCR for T. equi had county as significant (P<0.007) with hors-es in Zaria more likely to be positive and no association with sex (P>0.84), cELISA for T. equi was significant for both (P>0.001), where females and Zaria were more likely to be positive. Combining the two tests with 0 for all negatives, 1 for mixed results, and 2 for all positive, categorical analysis found females had mixed results (P<0.02), and horses from Zaria were likely to be positive (P<0.001). Theileria haneyi was not significant for sex (P>0.18) or county (P>0.70). Nested PCR of B. caballi was not significant for sex (P=1) or county (P>0.79). Species was not significant for all parasite detection methods (P>0.20), and there was no effect of county (P>0.34). Sex could not be tested as there were only males in Zaria, where most of the horses sampled had ticks. The number of ticks per horse was higher in Zaria for all testing methods (P<0.02) and strongly associated with T. equi nPCR (P<0.001) with positive animals having more ticks. Competitive ELISA of T. equi was not associated with tick number (P>0.46). Sex was not significant for the number of ticks for any testing method (P>0.77).”
- Why the authors failed to get positive results with cELISA for B. caballi for sample z-1, where nPCR test is positive? Too low prevalence of parasite in the blood for getting enough antibody for an ELISA test? Do the ELISA tests were run in duplicates to be sure about the result found for each sample?
Response: It is possible that the horse Z-1 was in the early infection, acute infection, where the humoral immune responses were just primed, and antibodies were not detectible by cELISA.
Lines 180-184: “During acute parasitemia, nPCR detects infection before an antibody response develops, yielding a negative cELISA result. In chronic infection, the low parasitemia stimulates antibody production for life. This again justifies the use of combined molecular and serological assays to overcome the constraint of single assays.”
- l.96 and Table 5: Did the authors test the presence of parasites (using the nPCR assay) in the ticks collected to check their infection status? It would have been interesting to see the prevalence of the different parasites in the different tick species and gender. This is particularly important to understand how the different parasite species are transmitted and which vector is used, or if a vector can carry different parasite species.
Response: In our understanding, ticks that are positive for the parasites does not mean it is a competent vector. During tick feeding, the parasites are ingested, and we would expect PCR positives. Now that we know which ticks are feeding on horses in Nigeria, the next experiments are to rear uninfected ticks in the laboratory and test if these ticks can acquire and transmit the parasite to naïve horses.
:
*Discussion:
- l.126-127: How long do the anti-T antibodies last in horse blood after an infection?
Response: We modified for clarification.
Line 181-182: “In chronic infection, the low parasitemia stimulates antibody production for life.”
- l.152-153: Could it be possible to chose very conserved or several conserved genomic regions for
each parasite species to be able to detect also parasite which harbor variation in these genomic regions?
This shouldn't be a problem if you have several molecular markers for each species, designed in conserved regions.
Response: All nPCR assays were used previously in several publications and validated in our lab. The primers used in the study were designed to conserved regions for each species known at the time of publication.
- l. 155-157: Why not having tested these collected ticks for the presence of parasites using the nPCR sets of primers? Could the authors identified which ticks were collected from which horse and add the info in their analysis. Authors could then potentially established correlations between presence of specific tick species and EP parasite in horse blood.
Response: In our understanding, ticks that are positive for the parasites does not mean it is a competent vector. During tick feeding, the parasites are ingested, and we would expect PCR positives. Now that we know which ticks are feeding on horses in Nigeria, the next experiments are to rear uninfected ticks in the laboratory and test if these ticks can acquire and transmit the parasite to naïve horses.
*Materials and Methods:
- Looking at the Table 1, we can see that we clearly have a sex ratio bias for each of the two breeds: could the authors explain why? With such bias, we cannot test for parasite prevalence by sex for each breed but for all the breeds together it is still possible and valuable.
Response: This study was not designed to compare gender associated with infection. Horses showing varying clinical signs suggestive of piroplasmosis were selected for this study irrespective of gender.
- l.175: Do RBC counts where performed using microscopy only? This is not very clear. Did the authors also spotted parasites while evaluating the level of RBC or never found any parasite?
How PCV was measured?
How hemoglobin rate was assessed?
Response: We modified the text for clarification.
Lines 242-244: Complete blood counts were performed for all 72 horses by the College of Veterinary Medicine, University of Abuja, Nigeria, using an automated hematology analyzer (Pioway Medical Lab Equipment Co., Ltd, China).
Line 91-95: “Before sample collection, the 72 horses examined in this study showed variable clinical signs. However, microscopic examination of Giemsa-stained thin blood smear did not detect any infected erythrocytes. Red blood cell (RBC) count, packed cell volume (PCV), and hemoglobin showed 41 horses were below the normal range for at least one out of the three parameters (Table 1, supplementary data 1).”
All these methodological information are missing and should be added to the manuscript.
Response: We modified the methodology.
Lines 242-244: Complete blood counts were performed for all 72 horses by the College of Veterinary Medicine, University of Abuja, Nigeria, using an automated hematology analyzer (Pioway Medical Lab Equipment Co., Ltd, China).
- l.191: How did the authors choose the targeted genes and what is the potential function of these genes in the EP parasites (if known)? How the primers were designed?
Response: The target genes for the nPCR are given in Table 2, and previous publications describing the development of these species-specific nPCRs. The ema1 and rap1 genes were chosen as the genes encoding the target antigens of the respective cELISAs. For T. haneyi, a gene encoding a hypothetical protein and occupies the space where ema1 does in the comparable location of the T. equi genome was used. Ema1 and rap1 encode membrane proteins with predicted roles in erythrocyte invasion. There is no predicted function for the hypothetical protein of T. haneyi.
Primers were chosen using the program Oligo by choosing conserved sequences with predicted efficient binding characteristics. The nPCR assays have been standardized, validated and used in several publications.
- l.197: Did the authors cleaned their PCR product after the first round PCR to deactivate the first set of primers? (using ExoSAP-it for e.g.).
Response: Nested PCR was performed per published protocols. No cleanup of the first reaction was done, and first reaction primers would be in the second reaction in small concentrations.
Lines 267-268: “ Second-round reactions contained 1 µl of the first-reaction product, 12.5 µl of PCR master mix (Roche, Indianapolis, IN, USA), and 11.5 µl of internal primer mix.”
There are also several information that are missing in this section, for example:
- How the authors separated their PCR products: Agarose gel? What % of agarose?
Response: We added requested information.
Lines 270-272: The PCR products were separated on 2% agarose gel (SeaKem LE, Lonza, Rockland, ME, USA) in 1X TAE (0.04 M Tris, 0.004 M acetate, 0.001 EDTA, pH 8.3).
- What are the electrophoresis conditions used in this experiment?
Response: We added requested information.
Lines 272-274: “Agarose gels were electrophoresed for 1 hour at 70 Volts and stained with SYBR Safe DNA stain according to manufacturer directions (ThermoFisher Scientific, Waltham, MA, USA).”
- What is the expected size of the amplicon for each pair of primers?
Response: We added the expected amplicon sizes to table 4.
Also could the authors provide a gel example of their PCR products to illustrate the kind of electrophoresis gel results for (i) a single infection for each of the EP parasites and (ii) a double infection?
Response: Because there have been agarose gels presented in previous publications that show the nature of the amplicons, we chose to not duplicate published data. The species-specific nPCRs were run individually, not as multiplexed reactions. Only single bands were seen in each lane of infected animals.
======= MINOR COMMENTS =======
------- COMMENTS ON THE TEXT -------
- Minor comments -
*Results:
- l.88: Please remove the blank space before "In addition" and check in the whole manuscript if more than one blank space is present at the beginning of sentences.
Response: We removed as recommended.
- l.89: Please be sure to always put all the scientific names in italic in the whole manuscript.
Response: We checked all scientific names.
*Materials and Methods:
- Authors have to check how they numbered their tables: Table 1 should be first and is coming at the end of the manuscript.
Response: We corrected table numbers.
----- COMMENTS ON THE REFERENCES -----
- l.60: The authors mentioned "all these studies" in the introduction but did not provide references for them. Could the authors add these particular refs in their present manuscript?
Response: We revised the statements.
Reviewer 3 Report
Please see the attached file.

Author Response
Dear Editor
We appreciate the comments and have responded point-by-point to each comment with changes to the manuscript.
Reviewer #3:
General comments: The purpose of this manuscript was to determine the molecular and seroprevalence of Equine Piroplasmosis (EP) Parasites infection in Nigerian horses using combination of nested PCR (nPCR) and competitive enzyme-linked immunosorbent assay (cELISA). This manuscript was described relatively well, and the results would be informative to detect the EP pathogens. However, several major points should be addressed and revised properly in the manuscript.
Minor comments: 1. Species name should be italic and also the genus name should be abbreviated. 2. Abbreviations should be spelled out in full at first; and after the first appearance, the abbreviation should be used. For example, PCV?, OIE?, nPCR?, cELISA?, FTA? 3. All numbers of tables should be rearranged. 4. Page 9, line 212: Please insert the appropriate reference for Walker et al. (2014). 5. If possible, please add surveyed areas on the map.
Response: We revised the manuscript as requested.
Major comments: 1. Table 3 (actually, Table 1) should to be remade for better understanding and readability. I would like to recommend the format as shown below. The authors can improve it. a. Only values NOT belonging to its normal range should be italicized based on the author’s comments. But sample No. 12, 15, 16, etc are italicized although showing normal condition for RBC. If possible, each cell can be shaded to show the abnormal values. b. Can the authors provide the evidence for the normal range? The references should be cited in the text and footnote below the result table.
Response: We modified as requested.
*Normal range: 6.2-10.2 for RBC [ref??]; 31-50 for PCV [ref??]; 11.4-17.3 for Hb
[ref??]. See the Section xxx for the detailed criteria.
Response: We added references for the normal range.
Harvey JW. Normal hematologic values. . In: Equine Clinical Neonatology (eds AM Koterba, WH Drummond & PC Kosch). 1990:561–70. .
Harvey RW, Asquith, R.L., McNulty, P.K., Kivipelto, J. & Bauer, J.E. Haematology of foals up to one year old. . Equine Vet J. 1984;16:347–53.
- Table43 (actually, Table 2) should to be improved for better understanding and
readability. I would like to recommend the format as shown below. The authors can
improve it further
Response: We modified as requested.
- One of the disadvantages of ELISA is that the presence of the antigen in the sample can indicate infection in the past, NOT in the present. Nonetheless, why do authors think that cELISA is important for diagnosis? Please describe it in the discussion.
Response: We modified the text for clarification.
Lines 190-193: Possible explanations for the lower nPCR detection are a low number of parasites in the peripheral blood of carrier state animals or horses having received treatment to eliminate or reduce parasitemia. Both scenarios may result in antibody detection with nPCR being negative as previously described [1].
cELISA was chosen because they are the official tests approved by OIE.
- Were ixodid ticks collected from the same horse samples? Sample ID need to be linked to the detailed information, such as horse breed, sampling spots, etc.
Response: We added supplementary Table 2 to the manuscript.
- Please add the limitation of this study epidemiologically and technically. For examples, there were limitation of sample numbers, various condition of samples (difficult to standardization of sample condition), and so on in the survey.
Response: The intent of this study was to define conditions necessary for further epidemiological studies. This work itself was not designed as an epidemiological study.
.
Reviewer 4 Report
The manuscript described detection of equine piroplasmosis parasites in Nigerian horses with using molecular and serological methods. The article is well written and informative.
However, few things should be improved/considered:
1.Introduction. In my opinion working hypothesis and objectives have not been clearly defined. Please consider to make it clearer for readers.
2.M&M. The authors provide information about clinical signs (line 167-169). Similar sentence is also in Results part (line 76-79). Please consider to focus on describing the symptoms in one of them.
3.M&M. The authors mentioned there are no Elisa tests for detecting specific antibody against T. haneyi. Recently, the manuscript of Bastos et al., 2021 was published. Thus you should add there is no „commercial”(line 207) or just remove this sentence.
4 .M&M. The authors reveals that it was „various breeds” whereas according to article it was only two.
5.Results and Discussion are generally ok.
6.There was a tendency to use the name of parasites without italic throughout the manuscript.
This is not the correct. Full parasites names should be italic.
7.In my opinion it would be worth to make minor English language editing to improve instances of awkward syntax.
Author Response
Dear Editor
We appreciate the comments and have responded point-by-point to each comment with changes to the manuscript.
Reviewer #4
The manuscript described detection of equine piroplasmosis parasites in Nigerian horses with using molecular and serological methods. The article is well written and informative.
However, few things should be improved/considered:
1.Introduction. In my opinion working hypothesis and objectives have not been clearly defined. Please consider to make it clearer for readers.
Response: we modified for clarification.
Lines 79-82: “Here, we report using the World Organization for Animal Health (OIE) approved cELISA serological tests [23] combined with molecular detection for T. equi, T. haneyi, and B. caballi to increase our ability to detect horses exposed to pathogens that cause EP.”
2.M&M. The authors provide information about clinical signs (line 167-169). Similar sentence is also in Results part (line 76-79). Please consider to focus on describing the symptoms in one of them.
Response: We removed duplicate information from the result section.
3.M&M. The authors mentioned there are no Elisa tests for detecting specific antibody against T. haneyi. Recently, the manuscript of Bastos et al., 2021 was published. Thus you should add there is no „commercial”(line 207) or just remove this sentence.
Response: We added “commercial” as recommended and the reference.
Bastos RG, Sears KP, Dinkel KD, Kappmeyer L, Ueti MW, Knowles DP, et al. Development of an Indirect ELISA to detect equine antibodies to Theileria haneyi. Pathogens. 2021;10 3; doi: 10.3390/pathogens10030270.
4 .M&M. The authors reveals that it was „various breeds” whereas according to article it was only two.
Response: We specified that two breeds were used in this study.
Lines 226-228: 4.1. “Study population. A purposive sampling technique using 72 horses of West African Barb or Argentine Polo Pony breeds and sexes (Figure 1) showing varying clinical signs suggestive of piroplasmosis was collected for this study.”
5.Results and Discussion are generally ok.
6.There was a tendency to use the name of parasites without italic throughout the manuscript.
This is not the correct. Full parasites names should be italic.
Response: We corrected all the scientific names.
7.In my opinion it would be worth to make minor English language editing to improve instances of awkward syntax.
Response: We checked using Grammarly to correct awkward syntax.
Round 2
Reviewer 1 Report
The revised manuscript is good.
Author Response
No comments.
Reviewer 2 Report
In this study, Dr. Idoko and colleagues have highlighted the importance of using a combination of molecular and serological test to improve the ability to detect equine piroplasmosis pathogens in horses in Nigeria. In their conclusions, the authors emphasized on the importance of conducting a continuous surveillance using these highly sensitive methods to understand parasite transmission and therefore, prevent spreading of EP parasites. Overall, the manuscript is well written and easy to read and understand and the data collected are highly valuable. After the first round of revision, the authors really improved their manuscript by providing more details regarding their protocols and valuable information regarding EP parasites transmission and symptoms occurring in horses. They also strongly improved the presentation of their data: they have refined tables as suggested and have included statistical analysis of their dataset.
However, there is still some unclear sentences (especially in the paragraph where authors described the statistical comparisons they have done (see below). This point (and few other typos, see below) should be corrected before the full acceptance of the manuscript for publication.
Decision: Minor revision
======= LEGENDS =======
p.: page
§: paragraph
l.: line
Fig.: figure
Tab.: table
> : replaced by
======= MAJOR COMMENTS =======
------- COMMENTS ON THE TEXT -------
- Major comments -
*Results:
-l. 133-144: This paragraph describing the statistical analysis done on the dataset is not clear written. Authors could modified as proposed: "Binomial....of nPCR for T.equi showed that horses coming from Zaria are significantly more infected (P<0.007) compared to horses coming from Igabi or Keffi. We did not found any association between EP infection and horse sex (P>0.84). cELISA results highlighted that mare coming from Zaria are significantly more positive to T. equi (P<0.001 - problem here with the sign: if the result it's significant, P<0.001 and not P>0.001 as it's written in the present manuscript) than other horses tested"
-l. 135: "females had mixed results". What does this mean? Could authors clarify this sentence? May I proposed: "With a p-value <0.02, female horses are significantly more infected with EP than male and
overall, horses from Zaria are significantly more infected (P<0.001) than horses from other locations sampled. No differences between horse sex or location were detected for T. haneyi (sex:p>0.18; location: p>0.70) or B. caballi (sex: p=1; location: p>0.79) parasites".
-l. 138: I suggest to rephrase the sentence: "We did not demonstrated significant differences in the sensitivity of the two detection methods employed in this study (i.e. nPCR and cELISA) for the 3 parasite species tested (p>0.20). Moreover, we did not detect any significant differences in term of parasite prevalence in the 3 tested locations (p>0.34)".
======= MINOR COMMENTS =======
------- COMMENTS ON THE TEXT -------
- Minor comments -
*Materials and Methods:
-l. 257: Please remove italic font for "Agarose gels were...".
-l. 268: Please remove the period at the beginning of the sentence (should be at the end of l.267).
-l. 274: Please remove the period at the beginning of the sentence (should be at the end of l.273).
Please specify the significance threshold for all your statistical analyses in the §Statistical analysis: "all p-value < 0.05 are considered significant".
----- COMMENTS ON THE REFERENCES -----
None
Author Response
Reviewer #2
- Major comments -
*Results:
-l. 133-144: This paragraph describing the statistical analysis done on the dataset is not clear written. Authors could modified as proposed: "Binomial....of nPCR for T.equi showed that horses coming from Zaria are significantly more infected (P<0.007) compared to horses coming from Igabi or Keffi. We did not found any association between EP infection and horse sex (P>0.84). cELISA results highlighted that mare coming from Zaria are significantly more positive to T. equi (P<0.001 - problem here with the sign: if the result it's significant, P<0.001 and not P>0.001 as it's written in the present manuscript) than other horses tested"
-l. 135: "females had mixed results". What does this mean? Could authors clarify this sentence? May I proposed: "With a p-value <0.02, female horses are significantly more infected with EP than male and
overall, horses from Zaria are significantly more infected (P<0.001) than horses from other locations sampled. No differences between horse sex or location were detected for T. haneyi (sex:p>0.18; location: p>0.70) or B. caballi (sex: p=1; location: p>0.79) parasites".
-l. 138: I suggest to rephrase the sentence: "We did not demonstrated significant differences in the sensitivity of the two detection methods employed in this study (i.e. nPCR and cELISA) for the 3 parasite species tested (p>0.20). Moreover, we did not detect any significant differences in term of parasite prevalence in the 3 tested locations (p>0.34)".
Response: We modified for clarification the statistical analysis results as recommended.
Line 141-160: “Binomial categorization of nPCR for T. equi showed that horses from Zaria were significantly more infected (P<0.007) compared to horses from Igabi or Keffi. We did not find any association between T. equi infection, as detected by nPCR, and horse sex (P>0.84). Competitive ELISA results highlighted that mares and horses from Zaria were significantly more positive for T. equi (P<0.001). Female horses were significantly more likely to be detected as infected with T. equi than males (P <0.02) when both testing methods were analyzed together, and overall, horses from Zaria were significantly more infected (P<0.001) than horses from other locations sampled. No differences between horse sex or location were detected for T. haneyi (sex: P>0.18; location: P>0.70) or B. caballi (sex: P=1; location: P>0.79) parasites. We did not demonstrate significant differences in two detection methods employed in this study (i.e., nPCR and cELISA) for the tick species identified on horses for parasite species tested (P>0.45). No ticks were found on horses that tested positive for T. haneyi. Moreover, we did not detect any significant differences in parasite prevalence in the three tested locations (P>0.34) when tick species were included in the model. Sex could not be tested as there were only males in Zaria, where most of the horses sampled had ticks. The number of ticks per horse was higher in Zaria for all testing methods (P<0.02) and strongly associated with T. equi nPCR (P<0.001) with positive animals having more ticks. Competitive ELISA of T. equi was not associated with tick number (P>0.46). Sex was not significant for the number of ticks for any testing method (P>0.77).”
======= MINOR COMMENTS =======
------- COMMENTS ON THE TEXT -------
- Minor comments -
*Materials and Methods:
-l. 257: Please remove italic font for "Agarose gels were...".
Response: In our version and PDF, “Agarose gels were..” is not in italic font.
-l. 268: Please remove the period at the beginning of the sentence (should be at the end of l.267).
Response: In our version there is no period that precede lines 267-268. They read: ”…at 4ËšC. Second-round reactions contained 1 µl of the first-reaction product, 12.5 µl of PCR master mix (Roche, Indianapolis, IN, USA), and 11.5 µl of internal primer mix.”
-l. 274: Please remove the period at the beginning of the sentence (should be at the end of l.273).
Response: In our version there is no period that precede line 272-274. They read: “...EDTA, pH 8.3). Agarose gels were electrophoresed for 1 hour at 70 Volts and stained with SYBR Safe DNA stain according to manufacturer directions (ThermoFisher Scientific, Waltham, MA, USA).”
Please specify the significance threshold for all your statistical analyses in the §Statistical analysis: "all p-value < 0.05 are considered significant".
Response: We added the statement in the materials and methods, statistical analysis section, as recommended.
